# Feline Gastrointestinal Eosinophilic Sclerosing Fibroplasia—Extracellular Matrix Proteins and TGF-β1 Immunoexpression

**DOI:** 10.3390/vetsci9060291

**Published:** 2022-06-13

**Authors:** Néstor Porras, Agustín Rebollada-Merino, Fernando Rodríguez-Franco, Andrés Calvo-Ibbitson, Antonio Rodríguez-Bertos

**Affiliations:** 1VISAVET Health Surveillance Centre, Complutense University of Madrid, 28040 Madrid, Spain; nestorpo@ucm.es (N.P.); agusrebo@ucm.es (A.R.-M.); 2Department of Animal Health, Faculty of Veterinary, Complutense University of Madrid, 28040 Madrid, Spain; 3Department of Internal Medicine and Animal Surgery, Faculty of Veterinary Medicine, Complutense University of Madrid, 28040 Madrid, Spain; ferdiges@ucm.es; 4VetPatólogos, Av. Isabel de Farnesio, 27, 28660 Boadilla del Monte, Spain; a.calvo@vetpatologos.com

**Keywords:** collagen, feline gastrointestinal eosinophilic sclerosing fibroplasia, fibronectin, intestinal fibrosis, immunohistochemistry, TGF-β1

## Abstract

Feline gastrointestinal eosinophilic sclerosing fibroplasia (FGESF) has been described as an inflammatory disorder with an eosinophilic component with etiopathogenesis that is still unknown. Sixteen intestinal samples from two veterinary diagnostic services (2014–2017) were included in the study. A histopathological criterion classified the cases into three grades (mild, moderate, and severe) according to the distribution of the lesions and the course. An immunohistochemical study of collagen I, collagen III, fibronectin, and transforming growth factor β1 (TGF-β1) was performed in each case. An immunohistochemical study of mild grades shows greater collagen III immunoexpression, compared to collagen I and fibronectin, which suggests an “early” stage of fibrosis. In more intense grades, an increased immunoexpression of collagen I, compared to collagen III, suggests a “late” stage of fibrosis. Otherwise, the highest expression of TGF-β1 was observed in the moderate phase, due to the high proliferation of reactive fibroblast and intense inflammation. The results suggest that the inflammatory infiltrate is the trigger for the elevation in TGF-β1, altering the collagen type III:I ratio. In conclusion, immunohistochemical studies can be a very useful method in diagnosing cases of FGESF of mild grades and could help to apply a differential diagnosis regarding feline eosinophilic chronic enteritis (CEE) in the context of inflammatory bowel disease (IBD).

## 1. Introduction

Feline gastrointestinal eosinophilic sclerosing fibroplasia (FGESF) has been described as part of the inflammatory diseases with an eosinophilic component described in the domestic cat, which include indolent ulcer, eosinophilic plaque, eosinophilic granuloma, and hypereosinophilic syndrome [1,2,3,4]. The etiopathogenesis of FGESF is not completely understood, partly because the majority of publications deal with single isolated cases [4,5,6,7,8,9].

There is the hypothesis that some cats with a genetic predisposition to this lesion develop eosinophilic inflammation in response to certain antigens in the intestinal environment [1]. Moreover, in the study by Linton et al., it was observed that long-haired cats and, specifically, the Ragdoll breed were overrepresented, suggesting a breed predisposition to this disease [2]. However, further studies concerning the genetic predisposition between affected cats are necessary to reach a conclusion.

Predisposing factors triggering exacerbated inflammation may include diet composition (food allergies or intolerances), intestinal microbiota dysbiosis, intracellular bacterial infection, extracellular endoparasites (*Cylicospirura* spp. in cougars–*Puma concolor*) [2,10], fungi, and hair or plant ingestion [6]. Some authors have reported similar eosinophilic sclerosing lesions in the subcutaneous tissue and abdomen of cats infected with methicillin-resistant *Staphylococcus aureus* [5,11].

Sex or age predisposition has not been described in association with FGESF. However, more than 70% of reported cases occurred in male cats [1]. This disease can affect cats of any age (1–16 years old), although is mostly diagnosed in adult cats aged 7–8 years [1,2]. Intestinal fibrosis causes stenosis and signs of obstruction. The main clinical manifestations in affected cats include weight loss, hyporexia, pale mucous membranes, vomiting, chronic diarrhea, and, on other occasions, constipation, as intestinal fibrosis causes stenosis and signs of obstruction [1,2]. Some cats may show acute onset with hematemesis, when the disease affects the pylorus [6]. A clinical survey reveals lesions characterized by a large, hard, easily palpable mass, most commonly found near the pylorus or ileocaecocolic junction [1,2]. Abdominal palpation usually confirms the presence of one or multiple, sometimes painful abdominal masses, which can be confirmed by complementary imaging techniques (ultrasound, radiology, or computed tomography) [3]. Ultrasonography in association with cytological examination is useful to focus the diagnosis of FGESF, although the definitive diagnosis is made through histopathology [3]. 

Grossly, FGESF has been roughly described as an intramural, firm, irregular, and ulcerated mass affecting the gastrointestinal tract, mainly in the stomach, pyloric sphincter, ileum, ileocecocolic junction, and colon [2,6]. The mesenteric lymph nodes may also be affected, with subsequent mesenteric lymphadenomegaly, suggesting an extension of the inflammatory process [6]. An atypical form of FGESF, characterized by small multifocal firm nodules affecting only the mesentery, has been also described [7]. Recently, a case was reported characterized by an extensive intramural cavitated mass in the cranial abdomen: the mass was attached to the pylorus and affected the proximal duodenum, the bile duct, and the pancreas, which penetrated the wall of the mass [9]. On rare occasions, the process initiated in the abdomen spreads to the thoracic cavity, probably due to drainage of the abdominal lymph nodes to the sternal lymph node, causing dyspnea [2].

Histologically, the key features of FGESF are dense trabeculae of collagen (in some cases resemble as osteoid tissue), interspersed by large spindle-shaped cells (fibroblasts), with accompanying mixed inflammatory cells (composed mainly by eosinophils with fewer lymphocytes, plasma cells, and, in some cases, mast cells) [1,2,5]. Due to its microscopic structure, FGESF can be misdiagnosed as lymphoblastic B-cell lymphoma, osteosarcoma, or sclerosing mast cell carcinoma, so pathologists must be extremely careful to distinguish this disease from a neoplasm [1]. The prognosis is variable and depends on factors such as diagnosis delay, involvement of adjacent organs, and response to therapy [3,8].

Eosinophilic inflammation is considered a key feature in this disease. It is considered that sclerosing fibrosis is boosted by the stimulation of fibrogenic mediators produced by eosinophils [1,2,4]. The cytoplasm of eosinophils is occupied mostly by granules, which contain four major classes of proteins. These proteins include major basic protein-1 and -2 (MBP-1, MBP-2), eosinophil peroxidase (EPX), eosinophil cationic protein (ECP), and eosinophil-derived neurotoxin (EDN) [12]. The major basic protein (MBP) is a cytotoxic peptide that stimulates the release of histamine and other inflammatory mediators [1,5,12]. In human beings, the deposition of MBP was present in lesions with inflammatory fibrosis, although it was absent in cases of non-inflammatory fibrosis [13]. Another study found elevated levels of MBP in the serum of some patients suffering from diffuse systemic cutaneous sclerosis [14].

Fibrogenic mediators, such as transforming growth factor β (TGF-β) and interleukin-1β (IL-1β), are produced by the activation of eosinophils and can lead to the proliferation of fibroblasts and extracellular matrix-proteins deposition [1]. An important inflammatory mediator in the process of initiating TGF-β production by eosinophils is leukotriene D4. Prednisone inhibits the production of leukotrienes by interrupting the arachidonic-acid route. Therefore, cats affected by FGESF and treated with prednisone showed the longest survival times [1].

Fibrosis is characterized by an alteration of collagen metabolism that leads to a greater deposition of this extracellular matrix protein in the tissue. There are several types of collagens, with type I and type III being the main components of the normal stroma. However, the proportions expressed in the tissues can vary significantly between tissues, both in physiological and pathological conditions [15,16,17]. Fibroblasts, among other cells, such as myofibroblasts and smooth muscle cells, are responsible for the deposition of collagen and scarring in numerous diseases [18,19,20,21]. Activated fibroblasts found in wounds during the healing process express high levels of fibronectin, procollagen, and smooth muscle actin [21]. The process of healing and deposition of collagen are regulated by multiple growth factors, including TGF-β1 and insulin-like growth factor (IGF), which accelerate healing [15,17,21]. However, the prolonged expression of this growth factors during chronic inflammation can induce excessive collagen deposition and eventually fibrosis [15,17].

TGF-β1 is chemotactic for inflammatory cells and fibroblasts; it stimulates the secretion of collagen by the smooth muscle cells (SMC) and fibroblasts, decreases the secretion of metalloproteinases (MMP), and increases the production of the tissue inhibitors of metalloproteinases (TIMP) [15,18]. TGF-β1 can be demonstrated in activated fibroblasts by means of in situ hybridization and immunohistochemistry, suggesting that the matrix associated with TGF-β1 could be a stimulus for the persistent expression of connective-tissue genes [15,17]. IGF-1 also stimulates the production of collagen by SMC and fibroblasts, while increasing cell proliferation and reducing the cellular expression of MMP-1, again leading to collagen deposition [13,22,23,24].

The aim of our study is to examine in depth the intrinsic mechanisms of intestinal fibrosis that are reflected in the abnormal extracellular matrix-protein deposition in the context of FGESF. We anticipate that the immunoexpression of extracellular matrix proteins (fibronectin, collagen I, and collagen III) and TGF-β1, which may play an important role in the development of this entity, could help to determine the disease grade and distribution in affected cats with a more accurate forecast. In addition, in this study we propose a classification of the lesion according to its severity and distribution.

## 2. Materials and Methods

### 2.1. Study Samples

Sixteen feline intestinal samples obtained from well-oriented endoscopic feline intestinal biopsies (*n* = 8) and full thickness (*n* = 8) diagnosed as FGESF, from two veterinary-diagnostic services (Veterinary Teaching Hospital, Faculty of Veterinary Medicine, Complutense University of Madrid and VetPatólogos diagnostic laboratory) during a 4-year period (2014–2017), were included in the study. All cases were retrospectively re-evaluated by two experienced veterinary pathologists and two trainee veterinary pathologists, with a final diagnosis of FGESF achieved by agreement (Table 1). Ethical review and approval were waived for this study due to the employment of samples previously used for diagnostic purposes.

The endoscopic biopsies correspond to animals in which, during the ultrasonographic and endoscopic study, only a thickening of the mucosal layer was observed, without a “mass effect”. These endoscopic samples were compatible with a chronic eosinophilic enteritis (CEE) and histologically were diagnosed as FGESF grade I in this study (discussed later). All animals contributing to this study were sampled from each small intestinal tract (duodenum, proximal jejunum, and ileum). Herein, we showed only the intestinal tracts with FGESF-compatible lesions (duodenum *n* = 14, jejunum *n* = 13, and ileum *n* = 10). 

### 2.2. Histological Processing

Tissue samples were fixed in formalin for 24 h, automatically processed (Citadel 2000 Tissue Processor, Thermo Fisher Scientific, Waltham, MA, USA), and embedded in paraffin (HistoStar Embedding Workstation, Thermo Fisher Scientific). Five consecutive sections of 4 µm thickness were obtained for each case using a microtome (FinesseMe+, Thermo Fisher Scientific). One section was stained with hematoxylin-eosin (Gemini AS Automated Slide Stainer, Thermo Fisher Scientific), and the following four sections were placed in positively charged glass slides and used for further immunohistochemical studies. 

### 2.3. FGESF Diagnosis and Evaluation

The histopathological features that must be fulfilled to diagnose a possible case of FGESF are as follows: fibrosis, reactive fibroblast infiltrate, and a mixed inflammatory infiltrate, mainly composed of a moderate to severe eosinophilic component, based on previously reported cases of FGESF [1,2,3,5,8]. The samples were reassessed according to the endoscopic, biopsy, and histopathological guidelines for the evaluation of gastrointestinal inflammation in companion animals, from the World Small Animal Veterinary Association (WSAVA), on a scale of 0–3 (0: absent, 1: mild, 2: moderate, and 3: severe) [25].

Furthermore, the eosinophilic inflammation was graded as mild (1: 5–10 eosinophils per 40× field), moderate (2: 10–20 eosinophils per 40× field) or severe (3: eosinophils predominate and are not easily enumerated), based on the study by Tucker et al., on CEE [26]. We have also evaluated the presence of reactive fibroblasts and the presence of foci of necrosis on a scale of 0–3. The degree of reactive fibroblasts infiltrate was graded as mild (1: 10–30 reactive fibroblasts per 40× field), moderate (2: 30–60 reactive fibroblasts per 40× field), or intense (3: >60 reactive fibroblasts). 

Thus, we evaluated the morphologic and inflammatory parameters detailed in Table 2 in the mucosa, submucosa, muscular, and serosa layers. 

### 2.4. FGESF Grading Criteria

The classification criteria for each proposed grade (I, II, or III) were as follows: involvement of intestinal layers, fibrosis (differentiating discontinuous collagen bands in the lamina propria in grade I and from dense collagen trabeculae in grade II and III), reactive fibroblasts, foci of necrosis, and eosinophilic inflammatory infiltrate (Table 3). A case was assigned to each grade when at least 4 out of the 5 classification criteria were met. Furthermore, to diagnose a grade I FGESF, eosinophilic inflammation should be included within the moderate or severe grade, as proposed by Tucker et al. [26].

### 2.5. Immunohistochemistry

The paraffin sections placed in positively charged glass slides were deparaffinised in xylene and rehydrated. Antigenic retrieval was carried out for 3 min in a pressure cooker using tris-sodium citrate buffer 0.01 mol/L, with pH 6 (Panreac Química S.L.U., Barcelona, Spain). Endogenous peroxidase was blocked by immersing the samples in a 3% hydrogen peroxide in methanol solution (Panreac Química S.L.U.). Then, a commercial kit was used in accordance with the instructions of the manufacturer (Novolink Polymer Detection System, Leica, Wetzlar, Germany). The slides were incubated overnight at 4 °C with the primary antibodies detailed in Table 4 (Abcam, Cambridge, UK). Only one antibody was used within the same reaction round. Positive and negative controls were included in each batch of slides. For negative controls, the primary antibody was omitted and substituted by tris-buffered saline. After detection using a 3,3′- diaminobenzidine tetrahydrochloride hydrate-based method, the samples were counter-stained with hematoxylin (Gemini AS Automated Slide Stainer, Thermo Fisher Scientific). 

Type I and type III collagen immunoreactions were assessed as follows: fibrotic tissue was evaluated in 5 adjacent non-overlapping fields under low-power field (LPF) magnification (40×), and the percentage of immunostained fibrotic tissue was determined. A score from 0 to 3 was assigned to each sample: grade 1 (mild): 0–30% immunopositive fibrotic tissue; grade 2 (moderate): 30–60% immunopositive fibrotic tissue; and grade 3 (intense): more than 60% immunopositive fibrotic tissue.

Fibronectin immunoreaction was assessed as follows: 100 fibroblasts were counted in 5 adjacent, non-overlapping fields under a high-power field (HPF) magnification (400×) (500 cells in total), and immunostained cells were expressed as a percentage of the cell count. TGF-β1 immunoreaction was assessed as follows: 100 fibroblasts/macrophages were counted in 5 adjacent, non-overlapping fields under a high-power field (HPF) magnification (400×) (500 cells in total), and immunostained cells were expressed as a percentage of the cell count. A score from 0 to 3 was assigned to each sample: grade 1 (mild): 0–30% immunopositive cells; grade 2 (moderate): 30–60% immunopositive cells; grade 3 (intense): more than 60% immunopositive cells.

## 3. Results

### 3.1. Epidemiology

According to the established histopathological criteria, most cats were classified as grade I (8/16, 50.0%), followed by grade II (6/16, 37.5%), and, to a lesser extent, grade III (2/16, 12.5%) (see above).

The most-represented breed in our study was European Shorthair (9/16, 56.3%), followed by Persian (3/16, 18.8%), Siamese (2/16, 12.5%), Chartreux (1/16, 6.2%), and Norwegian (1/16, 6.2%). The cats classified as grade I included European Shorthair (6/8, 75.0%), Siamese (1/8, 12.5%), and Norwegian (1/8, 12.5%); those classified as grade II were Persian (3/6, 50.0%), European Shorthair (2/6, 33.3%), and Chartreux (1/6, 16.7%); and those classified as grade III were European Shorthair (1/2, 50.0%) and Siamese (1/2, 50.0%).

Males and female were equally represented (8/16, 50.0% each). In grade I, there were more females (5/8, 62.5%) than males (3/8, 37.5%), whereas in grade II, there were more males (4/6, 66.7%) than females (2/6, 33.3%). In grade III, there were one male and one female (1/2, 50.0% each).

The mean age of the cats included in the study was 5 years (range: 2–16 years). The mean age of the cats included was 5 years in grade I (range: 2–13 years), 6 years in grade II (range: 3–15 years), and 12 years in grade III (range: 8–16 years).

### 3.2. Histopathological Results

When we consider the different parts of the intestine, in most cats the disease affected the duodenum (14/16, 87.5%), followed by the jejunum (13/16, 81.3%) and the ileum (10/16, 62.3%). In several cases, the entire small intestine was affected (10/16, 62.5%).

Half of the cases (8/16, 50.0%) were classified as grade I, which is histologically compatible with CEE. In 6/8 cases (75.0%), the disease affected the whole of the small intestine; in 1/8 cases (12.5%) there were lesions in the duodenum and jejunum; and in 1/8 cases (12.5%) there were lesions just in the duodenum. These cases were characterized by multifocal mucosal fibrosis (Figure 1). The morphological changes in grade I (CEE) cases included moderate villus stunting, mild to moderate distension of the crypts, and moderate lymphangiectasia. Minimally to mildly reactive fibroblasts and multifocal areas of fibrosis of the lamina propria were observed. Such a lesion was characterized by the formation of a discontinuous band of fibrous connective tissue immediately beneath the villi, producing a marked separation of the villus–crypt junction, displacing the glandular crypts down from the lamina propria (Figure 1a,b). There was an absence of epithelial injury and necrosis. The morphological diagnosis in these cases was of moderate, chronic, diffuse, eosinophilic. and lymphoplasmacytic enteritis affecting one intestinal layer (mainly the lamina propria in the mucosa). Neutrophils were minimal, and macrophages and intraepithelial lymphocytes were scarce.

Six out of sixteen cases (37.5%) were classified as grade II. Only 2/6 cases (33.3%) involved the whole of the small intestine, 2/6 cases (33.3%) had lesions only in the duodenum, and 2/6 cases (33.3%) had lesions only in the jejunum. These cases were characterized by a moderate fibrosis and the presence of abundant fibroblasts organized in a continuous band of fibrous connective tissue (Figure 2). There was occasional mild to moderate epithelial injury in some cases and an absence of villus stunting, crypts distension, lymphangiectasia, and necrosis. The inflammatory response in these cases was mainly composed of a moderate eosinophilic and lymphocytic infiltrate, with the occasional presence of macrophages and plasma cells. Neutrophils and intraepithelial lymphocytes were scarce.

Grade III cases were underrepresented (2/16, 12.5%), where 2/2 cases (100%) involved the whole of the small intestine. These cases were characterized by the severe disruption of the intestinal-wall structure affecting mucosa (lamina propria), submucosa, muscularis, and, rarely, the serosa (intramural process); this was caused by abundant fibrosis and a moderate presence of reactive fibroblasts (Figure 3). There were slight multifocal areas of necrosis in both cases and moderate epithelial injury in one case. The rest of the morphological parameters were unchanged. The inflammatory infiltrate was composed of abundant eosinophils, lymphocytes, and macrophages in the lamina propria, submucosa, muscular, and serosa layer in one case. Plasma cell and neutrophilic infiltrate was observed inconsistently.

The histopathological assessment for each case is detailed in Table 5 and Table 6.

### 3.3. Immunohistochemistry

In grade I (CEE), the immunohistochemical study revealed moderate fibrillar immunoexpression for collagen I in the lamina propria, in the base of the villi, and above the crypts. Collagen III was expressed intensely and diffusely along the entire lamina propria, from the apical part of the villus to the muscularis mucosae. Fibronectin showed moderate immunoexpression in the base of the villi in the lamina propria, without affecting deeper areas. The expression of the three antibodies was intensely increased in cases with the characteristic fibrosis band. Regarding TGF-β1 expression, there was moderate cytoplasmic immunoexpression in the mononuclear cells and the epithelial crypt cells. In some cases, where fibrosis is predominant, there was slight staining in the loose connective tissue, due to the TGF-β1 protein binding to cellular membranes and the ECM located in the lamina propria (Figure 1).

In grade II, there was moderate immunoexpression for collagen I in the fibrotic areas located in the submucosa and the muscular interstitium. Collagen III was expressed moderately in the fibrotic areas as well as around the fibroblasts in the submucosa and the muscular layers. Fibronectin showed moderate immunoexpression around the fibroblasts in the submucosa and muscularis. TGF-β1 showed intense immunoexpression in the cytoplasm of the fibroblasts and macrophages in the submucosa and muscularis (Figure 2).

In grade III, there was intense immunoexpression for collagen I in the fibrotic bands sclerosing the submucosa, muscularis, and serosa. Collagen III was only immunoexpressed in perivascular areas, showing negative immunoexpression in the fibrotic bands. Fibronectin was expressed intensely around the fibroblasts. There was moderate TGF-β1 cytoplasmic immunoexpression in the fibroblasts and macrophages (Figure 3).

## 4. Discussion

FGESF is a newly recognized condition in feline medicine, with numerous unknowns surrounding etiology and disease progression. The few investigations have focused on case descriptions, diagnosis, and disease manifestation. In the Linton et al., study, it was observed that long-haired cats and, specifically, the Ragdoll breed were overrepresented, suggesting a breed predisposition to this disease [2]. Here, we reported the European Shorthair as the most commonly affected breed during a 4-year period. However, in this study, as in Craig’s research, no racial predisposition was observed, since the European Shorthair is representative of the feline population [1]. This suggests that breed distribution may be influenced by geographical location, given the trends in breeds among owners in different countries. Therefore, for an evaluation of a breed predisposition and breed-related severity of lesions, wider international multicentric studies are required. Although predisposition according to sex has been described [1], in our investigation we observed the same proportion of male and female cases. In our study, there were milder grade (I) cases among females, while there were a greater number of males with moderate grade (II) than females. The cases described in the literature comprise a wide age range, in agreement with our findings. However, the mean age in our study was at least 2 years below that of previous studies involving a significant number of cases [1,2]. Nonetheless, our findings confirm that most cats diagnosed with FGESF are middle-aged, suggesting the disease makes its clinical debut in this age group [2]. Interestingly, our cases with more advanced lesions showed a higher age range (8–16 years). However, a larger population is required to confirm, by statistical study, that there is a relationship between the development of advanced degrees of FGESF and older ages. 

Regarding the anatomical location of FGESF, the pyloric sphincter and the ileocecal junction are known to be involved in most cases [1,2]. By contrast, in our study, we observed no clear trend in localization at the level of the small intestine. In the histopathological study, we observed that in grade I, 6/8 cases (75.0%) involved the whole of the small intestine, while in grade II, only two cases showed generalized lesions, observing that 4/6 cases presented an isolated lesion in a single tract (duodenum or jejunum). Finally, in grade III, generalized lesions were again seen throughout the small intestine. This suggests that there may be an expansion of lesions over time. For the intermediate grade, it would be interesting to histologically analyze the remaining intestinal segments in order to confirm whether there is an adjacent inflammatory reaction typical of grade I or CEE. In the study by Tucker et al., two cats diagnosed with FGESF and transmural eosinophilic infiltrates at the site of the mass lesion also presented simple smooth-muscle hypertrophy, typical of CEE, in jejunal biopsies distant from the mass lesions. This study suggests that the histopathologic diagnosis may differ in different areas of the intestinal tract [26]. These results contrast with the study by Linton et al., who did not observe any cases of CEE adjacent to the specific lesion of the disease [2]. Further studies should include an increased number of cases, applying homogeneous evaluation criteria. This method would include the histopathological study of the different sections of the intestine and apply the grading proposed here, to determine if there is an extension of the fibrotic and inflammatory process that could correspond to an initial grade of FGESF. 

Histopathology is the gold-standard technique for the diagnosis of FGESF. As many aspects of its pathogenesis are still not fully understood, we proposed histopathological criteria for both the diagnosis and classification into grades of FGESF lesions in cats, which could serve future studies on disease outcome, prognosis, and therapy. Moreover, this could be useful, in the future, to differentiate a mild degree (grade I) of this disease (where the lesion is located exclusively in the mucosal layer) from CEE, with a prognosis and evolution that is more favorable, or, on the contrary, to determine if FGESF is the result of an evolution of CEE. We established three grades of FGESF, according to certain histopathological criteria but with a special emphasis on intestinal fibrosis. The mild grade (grade I) of FGESF, which is also compatible with CEE, featured a characteristic discontinuous band of fibrous connective tissue immediately beyond the villi in the villus-crypt junction of the lamina propria. This fibrotic-band deposition produced a marked separation between the basal part of the villi and the intestinal crypts. Mild to moderate morphological changes observed only during this mild grade, such as villus stunting, crypt distension, and lymphangiectasia, could also be evidence of a subacute to chronic intestinal response to injury.

The limitations of performing a histologic study of grade I FGESF only on endoscopic samples should be mentioned. Although during the ultrasound and endoscopic studies no lesion with a “mass effect” was observed, we cannot confirm that there was no involvement of deeper layers (submucosal, muscular, or serous). Therefore, further studies with full-thickness samples would be recommended to confirm our hypothesis.

The histological diagnosis of CEE in a cat with chronic gastrointestinal signs and no evidence of a predisposing cause can serve as an indicator for several variants of idiopathic eosinophilic GI disease, including diffuse idiopathic EE, FGESF, or hypereosinophilic syndrome. Currently, it is not known whether FGESF is initiated by the same process responsible for the development of the more common diffuse idiopathic CEE, although both diseases appear to have an underlying immune disorder [26]. It is challenging to differentiate CEE from grade I FGESF, so FGESF should be confirmed by histopathology when the disease progresses, as the deeper layers are affected. CEE in cats is an uncommon disease characterized by intramural-eosinophil infiltrates, where there may be hypertrophy of the gastric antrum and small-intestinal-muscle layers [26,27,28,29]. Moreover, additional histopathological changes for the intestine included mucosal fibrosis with a haphazard and multifocal pattern. However, this fibrotic lesion is occasional and uncharacteristic, differing from what has been previously observed and described in cases of Grade I with a discontinuous fibrotic band. Therefore, it is postulated that cases of CEE that present such a band of fibrosis at the junction between the villi and crypts may be most indicative of mild or early FGESF.

However, an immunohistochemical study can be a useful method to differentiate between the two processes. Minimally to mildly reactive fibroblast infiltration and more extensive collagen III immunoexpression, compared to collagen I and fibronectin, suggests an “early” stage of fibrosis [15,30,31,32,33]. Finally, we observed TGF-β1 still inside the cytoplasm of mononuclear cells and, to a lesser extent, in the crypt epithelial cells, which indicates that it has not yet been secreted. In some cases of fibrosis, there was a slight staining in the loose connective tissue due to the TGF-β1 protein binding to the cellular membranes and the ECM located in the lamina propria. These results suggest that chronic inflammation influences the expression of TGF-β1, causing a change in the deposition of the ECM due to an alteration in the collagen type III:I ratio [15,17]. The marked expression of these antibodies, in cases with the characteristic fibrosis band beneath the villi, reinforces the theory that this study can help in the differential diagnosis between CEE and FGESF. Moreover, the immunohistochemical results observed in this study are consistent with the conclusions reached in research by other authors studying the activity of extracellular matrix proteins during a fibrotic process in human inflammatory bowel disease (IBD), where the “early” stage of fibrosis is defined as the excessive deposition of collagen characterized by an increase in the accumulation of type III collagen, relative to type I collagen [15]. This type of fibrosis is expressed macroscopically as a mucosal thickening.

Grade II FGESF were reported to have two or more intestinal layers affected. The hallmark finding in this grade was the marked presence of an activated fibroblast population located between the bands of connective tissue in the lamina propria, in the submucosa, and, less commonly, in the muscularis. This grade shows the beginning of greater immunoexpression for collagen type I, compared to collagen type III, in the connective tissue. Fibronectin is still moderately expressed in the fibroblasts, as well as TGF-β1, which was intensely expressed in the cytoplasm of the activated fibroblasts and macrophages, due to phagocytosis. This suggests a progression of the disease with an equalization in the collagen type III:I ratio. At this point, there is a maximum expression of TGF-β1, where the degradation mechanisms of the ECM have diminished, and there is a continuous deposition of collagen and fibronectin. This may be related to the moderate inflammatory infiltrate present, stimulating TGF-β1 expression [15,17,21]. In this grade, due to the formation of dense collagen, the characteristic “mass effect” that we observe macroscopically begins to form.

The classification criteria for grade III FGESF cases included massive connective-tissue deposition, in at least three intestinal layers (lamina propria of the mucosa, submucosa, and muscularis layers). The chronic inflammation and activated fibroblast population are reduced. There is a maximum expression of collagen type I in the fibrotic bands, unlike collagen type III, which is only expressed perivascularly. Fibronectin was intensely expressed around the fibroblasts, as was TGF-β1, remaining in the cytoplasm of the narrow population of the fibroblasts and macrophages. These results are consistent with previous studies, which explain that the “late” stage of fibrosis is characterized by a decrease in active collagen deposition and a decrease in the ratio of collagen III:I (imbalanced fibrotic process) [15,31,32,33]. This late fibrosis stage corresponds to the extensive development of the characteristic “mass effect”, observed macroscopically.

The alteration of the proportion of type III:I collagen as well as the high expression of the protein and mRNA of TGF-β1 and IGF-1 coincide with the presence of an inflammatory infiltrate in the intestine of human individuals suffering from IBD [15]. Thus, it is proposed that the inflammatory infiltrate is the trigger for the increase in TGF- β1 and IGF-1 expression, inducing a phenotypic change in the fibroblast. However, these authors defend the idea that the findings are independent of the type of IBD; that is, they depend mainly on the presence and location of inflammation, but not on the type of inflammatory cells involved in the process [15]. These results suggest the need to differentiate between FGESF and IBD, since previous studies agree that the presence of eosinophils plays a very important role in the development of intestinal fibrosis in FGESF [1,2,5].

On the other hand, in the progression of the disease, the inflammatory process has only a secondary role, while fibrosis can progress in a self-perpetuating manner [21]. The continued expression of EMC proteins is due to the fact that TGF-β1 induces a persistent alteration in the regulation of collagen and fibronectin biosynthesis in the fibroblasts. It has been thought, in certain studies, that the continued increase in the rate of collagen production, even after removal of TGF-β1, is the result of the greater stability of newly transcribed collagen and fibronectin mRNAs. This could be due to the induction or stimulation of the synthesis of a protein or proteins capable of exerting a protective effect from the breakdown of newly synthesized transcripts [17,34]. It is possible that the determination of tissues destined for fibrosis occurs during the initial inflammatory process, where the levels of TGF-β1 and the proportion of type III:I collagen are determined [15,17,21]. The mechanisms that regulate fibrosis, therefore, appear to be distinct from those that regulate the inflammatory process. This explains the lack of efficacy of anti-inflammatory treatments in IBD, in preventing the evolution of fibrosis, once the collagen-deposition process has begun [21].

## 5. Conclusions

Intestinal fibrosis is a biologically heterogeneous process, involving multiple complex and interacting mechanisms. Our results suggest that collagen III is an important indicator of active fibrosis in postulated grade I FGESF. Collagen III expression could also be of diagnostic importance because grade I FGESF can be initially misdiagnosed as CEE, with a prognosis and evolution that is more favorable. In addition, our results also suggest an important role for TGF-β1, triggering fibroblastic proliferation as well as collagen and fibronectin deposition. Prospective studies, including a follow-up of animals diagnosed with grade I FGESF, would be necessary to confirm/discard the transition from CEE to FGESF. Future research regarding FGESF prognosis is needed to correlate clinical outcome with its pathogenesis. Studying biomarkers of FGESF will not only be of value in the treatment and management of patients but may also help to answer some of the questions regarding the pathogenic mechanism of intestinal fibrosis. 

## Figures and Tables

**Figure 1 vetsci-09-00291-f001:**
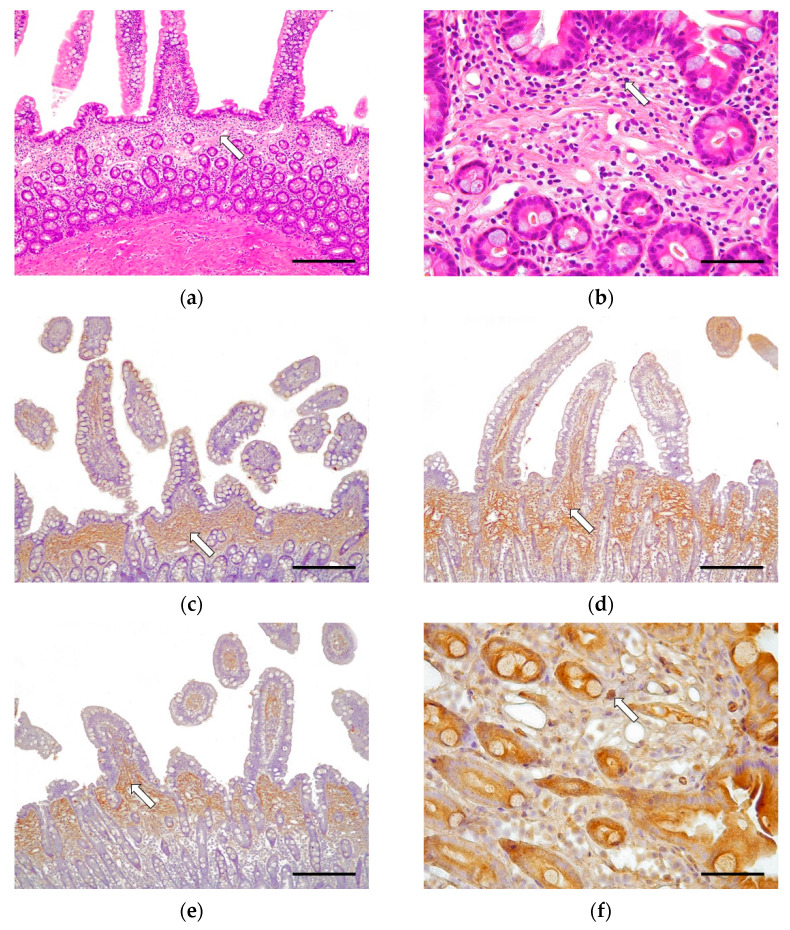
Chronic eosinophilic enteritis (Grade I). Cat, duodenum, case No. 4. (**a**) There is mild fibrosis that displaces down the glandular crypts (arrow). Hematoxylin–eosin. Scale bar: 1000 μm. (**b**) Surrounding the fibrotic areas, there is a moderate inflammatory infiltrate composed by lymphocytes, plasma cells, and eosinophils (arrow). Hematoxylin–eosin. Scale bar: 100 μm. (**c**) Moderate fibrillar immunoexpression for collagen I in the lamina propria, in the base of the villi, and above the crypts (arrow). Rabbit polyclonal anti-collagen I. Scale bar: 1000 μm. (**d**) Intense diffuse and reticullar collagen III immunoexpression in the connective tissue of the lamina propria (arrow). Rabbit polyclonal anti-collagen III. Scale bar: 1000 μm. (**e**) Moderate reticular fibronectin immunoexpression at the base of the villi in the lamina propria (arrow). Rabbit polyclonal anti-fibronectin. Scale bar: 1000 μm. (**f**) Moderate cytoplasmic TGF-β1 immunoexpression in epithelial cells of the crypts, mononuclear cells (arrow), and, to a lesser extent, in the lamina propria around fibroblasts. Rabbit polyclonal anti-TGF-β1. Scale bar: 100 μm.

**Figure 2 vetsci-09-00291-f002:**
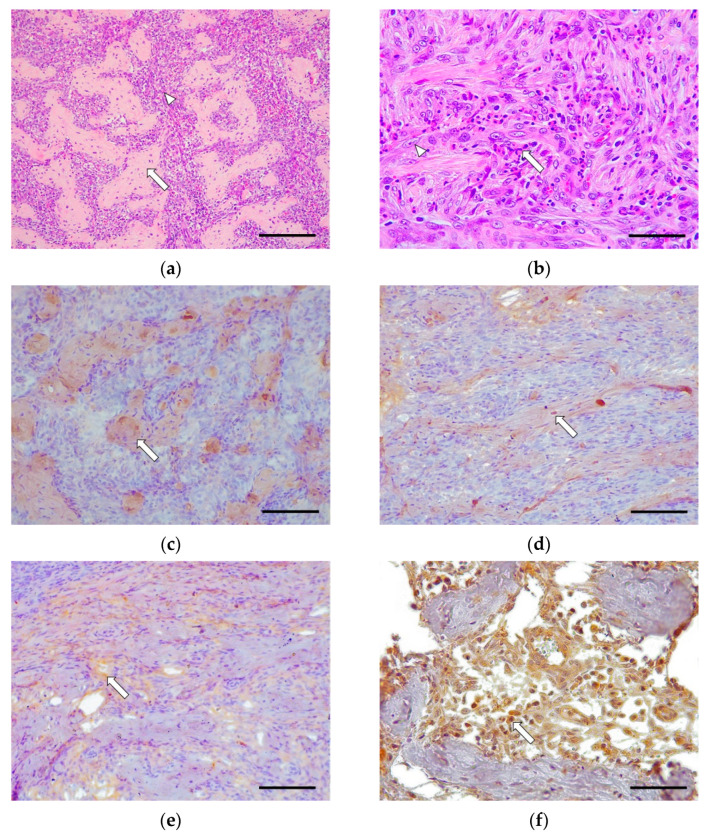
Feline Eosinophilic Gastrointestinal Sclerosing Fibroplasia (Grade II). Cat, duodenum, case No. 13. (**a**) Abundant collagen bands (arrow) surround a large number of proliferating fibroblasts and eosinophils (arrowhead). Hematoxylin–eosin. Scale bar: 400 μm (**b**) Large number of proliferating fibroblasts (arrowhead) and eosinophils (arrow). Hematoxylin–eosin. Scale bar: 100 μm. (**c**) Moderate fibrillar collagen I immunoexpression is observed in the collagen bands large number of proliferating fibroblasts and eosinophils (arrow), located at the level of the submucosa and muscular layers. Rabbit polyclonal anti-collagen I. Scale bar: 200 μm. (**d**) Moderate collagen III immunoexpression in the collagen bands and around the fibroblasts (arrow), located in the submucosa and the muscular layers. Rabbit polyclonal anti-collagen III. Scale bar: 200 μm. (**e**) Moderate diffuse immunoexpression is observed around proliferating fibroblasts (arrow) in the submucosa and the muscular layers. Rabbit polyclonal anti-fibronectin. Scale bar: 200 μm. (**f**) Intense diffuse TGF-β1 cytoplasmic immunoexpression in fibroblasts and mononuclear cells between the collagen bands (arrow), in the submucosa, and muscular layers. Rabbit polyclonal anti-TGF-β1. Scale bar: 100 μm.

**Figure 3 vetsci-09-00291-f003:**
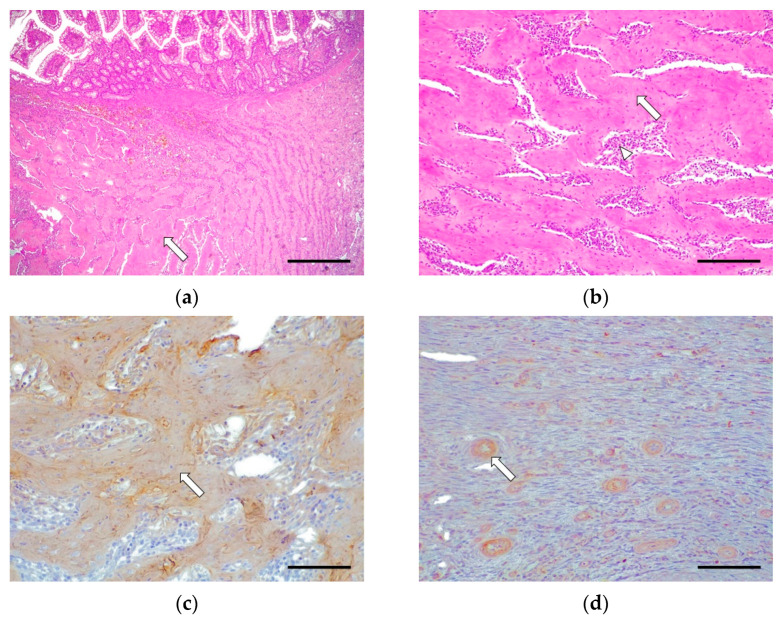
Feline Eosinophilic Gastrointestinal Sclerosing Fibroplasia (Grade III). Cat, du4odenum, case No. 16. (**a**). Dense collagen bands (arrow) occupying the lamina propria, submucosa, and muscular layer. Hematoxylin–eosin. Scale bar: 400 μm. (**b**) These bands (arrow) are surrounded by a mild to moderate number of fibroblasts and eosinophils (arrowhead). Hematoxylin–eosin. Scale bar: 100 μm. (**c**) Intense fibrillar collagen I immunoexpression in the dense bands (arrow) in the submucosa, muscular, and serosa layers. Rabbit polyclonal anti-collagen I. Scale bar: 200 μm. (**d**) Negative immunoexpression against collagen III is observed at the fibrotic bands, immunoexpressed only at the perivascular level (arrow) without affecting the sclerosing connective tissue. Rabbit polyclonal anti-collagen III. Scale bar: 200 μm. (**e**). Intense fibronectin immunoexpression in fibroblasts (arrow) in the submucosa and muscular layers. Rabbit polyclonal anti-fibronectin. Scale bar: 200 μm. (**f**). Moderate cytoplasmic TGF-β1 immunoexpression in fibroblasts and mononuclear cells (arrow), between the collagen bands as well as in the submucosa and muscularis. Rabbit polyclonal anti-TGF-β1. Scale bar: 100 μm.

**Table 1 vetsci-09-00291-t001:** Cases included in the study.

Case No.	Breed	Sex	Age (Years)	Affected Intestinal Tract
1	European Shorthair	Female	13	Duodenum
2	Siamese	Female	5	Duodenum, jejunum, ileum
3	European Shorthair	Male	13	Duodenum, jejunum, ileum
4	Norwegian	Male	4	Duodenum, jejunum, ileum
5	European Shorthair	Female	12	Duodenum, jejunum
6	European Shorthair	Female	2	Duodenum, jejunum, ileum
7	European Shorthair	Male	5	Duodenum, jejunum, ileum
8	European Shorthair	Female	2	Duodenum, jejunum, ileum
9	European Shorthair	Female	5	Jejunum
10	Persian	Male	7	Duodenum, jejunum, ileum
11	Chartreux	Male	3	Duodenum
12	European Shorthair	Male	14	Jejunum
13	Persian	Female	3	Duodenum
14	Persian	Male	15	Duodenum, jejunum, ileum
15	European Shorthair	Female	8	Duodenum, jejunum, ileum
16	Siamese	Male	16	Duodenum, jejunum, ileum

**Table 2 vetsci-09-00291-t002:** Morphological and inflammatory parameters evaluated in the cases included.

Morphologic Parameters	Inflammatory Parameters
Villus stunting	Intraepithelial lymphocytes
Epithelial injury	Lymphocytes
Crypt distension	Plasma cells
Lymphangiectasia	Eosinophils
Fibrosis	Neutrophils
Reactive fibroblasts	Macrophages
Foci of necrosis	

**Table 3 vetsci-09-00291-t003:** Histological criteria for classifying each evolutionary grade of FGESF.

Grade	Criteria
I	One intestinal layer affected (mucosa)
	Mild fibrosis
	Minimally to mildly reactive fibroblast infiltration
	Absence of foci of necrosis
	Moderate/severe eosinophilic inflammatory infiltrate
	(≥10 eosinophils per 40× field)
II	Two intestinal layers affected (mucosa/submucosa)
	Moderate fibrosis
	Severe reactive fibroblast infiltration
	Absence of foci of necrosis
	Eosinophilic inflammatory infiltrate
III	Three or more intestinal layers affected/Intestinal mural affected
	Severe fibrosis
	Moderate reactive fibroblast infiltration
	Presence of foci of necrosis
	Eosinophilic inflammatory infiltrate

**Table 4 vetsci-09-00291-t004:** List of antibodies used in the study.

Antibody	Type	Host	Dilution
Anti-Collagen I	Polyclonal	Rabbit	1:100
Anti-Collagen III	Polyclonal	Rabbit	1:100
Anti-Fibronectin	Polyclonal	Rabbit	1:200
Anti-TGF-β1	Polyclonal	Rabbit	1:200

**Table 5 vetsci-09-00291-t005:** Results of the histopathological evaluation and stage. Relation between the morphologic parameters with the affected layers and the grade of FGESF.

Case No.	Affected Intestinal Tract *	Affected Intestinal Layers	Villous Atrophy	Epithelial Lesion	Dilation of the Crypts	Lymphangiectasia	Reactive Fibroblasts	Fibrosis	Necrosis	Grade
1	D	Lp	++	+	++	+	+	+	−	I
2	D, J, I	Lp	++	−	−	++	+	+	−	I
3	D, J, I	Lp	++	−	+	++	+	+	−	I
4	D, J, I	Lp	++	−	+	++	+	+	−	I
5	D, J	Lp	++	−	+	++	+	+	−	I
6	D, J, I	Lp	++	−	+	++	+	+	−	I
7	D, J, I	Lp	++	−	+	++	+	+	−	I
8	D, J, I	Lp	−	−	−	++	+	+	−	I
9	J	Lp, Sb	−	−	−	−	+++	++	−	II
10	D, J, I	Lp, Sb	−	++	−	−	+++	++	−	II
11	D	Lp, Sb	−	+	−	−	+++	++	−	II
12	J	Lp, Sb	−	−	−	−	+++	++	−	II
13	D	Lp, Sb	−	++	−	−	+++	++	−	II
14	D, J, I	Lp, Sb, M	−	−	−	−	+++	++	−	II
15	D, J, I	Sb, M	−	−	−	−	++	+++	+	III
16	D, J, I	Lp, Sb, M, S	−	++	−	−	++	+++	+	III

* D, duodenum; J, jejunum; I, ileum; −, absent; +, mild; ++, moderate; +++, marked; * Lp, lamina propria; Sb, submucosa; M, muscular layer; S, serosa; I, mild grade (EE); II, moderate grade; III, severe grade.

**Table 6 vetsci-09-00291-t006:** Results of the histopathological evaluation and grade. Relation between inflammatory parameters with the affected layers and the grade of FGESF.

Case No.	Affected Intestinal Tract *	Affected Intestinal Layers **	Inflamatory Cells Infiltration						Grade
			IEL	Lymphocytes	Plasma Cells	Neutrophils	Eosinophils	Macrophages	
1	D	Lp	+	++	++	+	++	−	I
2	D, J, I	Lp	−	++	++	+	++	−	I
3	D, J, I	Lp	−	++	++	+	++	−	I
4	D, J, I	Lp	−	++	+	+	++	−	I
5	D, J	Lp	−	++	++	+	++	−	I
6	D, J, I	Lp	−	++	++	+	++	−	I
7	D, J, I	Lp	−	+	++	+	+++	−	I
8	D, J, I	Lp	+	++	++	−	++	−	I
9	J	Lp, Sb	−	+	−	−	++	++	II
10	D, J, I	Lp, Sb	−	++	++	+	+++	++	II
11	D	Lp, Sb	−	++	++	++	+	−	II
12	J	Lp, Sb	−	++	−	−	+++	++	II
13	D	Lp, Sb	−	++	−	−	++	++	II
14	D, J, I	Lp/Sb, M	−/−	+/++	++/−	−/−	−/++	−/++	II
15	D, J, I	Lp/Sb, M	−/−	++/++	++/−	−/+++	−/+	−/++	III
16	D, J, I	Lp, SB, M, S	−	++	−	++	+	++	III

* D, duodenum; J, jejunum; I, ileum; IEL, intraepithelial lymphocytes; −, absent; +, mild; ++, moderate; +++, marked; Lp, lamina propria; Sb, submucosa; M, muscular layer; S, serosa; I, mild grade (EE); II, moderate grade; III, severe grade. ** The injuries observed in the lamina propia are separated from the lesions in the submucosa and muscular layer by an “/”.

## Data Availability

Data are contained within the article.

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
