# Peer review of "Feline Gastrointestinal Eosinophilic Sclerosing Fibroplasia—Extracellular Matrix Proteins and TGF-β1 Immunoexpression"

_vetsci, 2022, doi:10.3390/vetsci9060291_

Round 1

Reviewer 1 Report

General comments: This manuscript provides some novel information regarding an uncommon cause of feline chronic enteropathies, particularly in the histopath characteristics. However, substantial editing is required to the reported methods and results to clearly evaluate the study’s conclusions. Care should also be taken not to extrapolate possible clinical applications to a retrospective, histopath-based study. Lastly, many comparative statements were made without statistical analysis; statistical analysis should be performed if these conclusions remain in the revised manuscript.

There are multiple areas where grammar requires revision. While I have highlighted some, this should be a general focus of the revision process.

Abstract: The abstract will require revision following other recommended edits to the manuscript (outlined below) to clarify the diagnostic methods applied to the study population.

Introduction:

Line 40: Is “Davidson et al., 2021” the same as reference 9 or a different citation?

Lines 42-43: While there has been some suggestion that Ragdoll’s are predisposed to FGESF, it is unclear how that relates to the previous sentence about cats in general been predisposed to an eosinophilic response. Evidence of Ragdoll predisposition to other eosinophilic diseases might more clearly connect these thoughts.

Lines 47-48: Please revise for clarity. Lesions similar to what?

Line 58: “Ileum” instead of “ileus”?

Line 65: “Penetrated” instead of “penetrate”

Lines 50-81: Reorganization of these paragraphs could help improve clarity. Intermixed description of examination findings, imaging findings, and cytologic/histopathologic features make this section difficult to follow. Consider grouping all PE findings, then imaging, then pathologic findings together.

Line 75: Consider removing “an”

Line 101: Does “this” refer to collagen?

Lines 101 - 104: Please divide into separate sentences for clarity (“There are several types of…in which the tissue is found”)

Line 103: Please expand/clarify what is meant by “depending on the state in which the tissue if found.” Does this mean stage of disease progression or stage of fibrogenesis?

Line 110: What is “its” referring to?

Lines 126-130: This seems more appropriate for the discussion, as this study is not designed to evaluate progression of disease or whether FGESF is a progression or separate disease process from CEE.

Materials and Methods

Lines 132-133: It is unclear how many samples were full thickness samples and how many were endoscopic. For the authors’ aims and suggested histopathologic classification, it does not seem appropriate to include cases in which full thickness samples were not obtained. Endoscopic sampling inherently does not allow assessment of all layers of the GI tract, which is an assumption of the grading system. Inclusion of cases with only endoscopic samples is a major limitation to the study’s results and conclusions and should be removed.

Line 136: The methods should include a description of diagnostic criteria for FGESF. It is unclear whether cases were screened with additional stains for neoplasia or infectious disease or degree of eosinophilic inflammation that was required to categories these cats as having eosinophilic inflammation. This is critical to understand the study population, as MCT have been previously misdiagnosed as eosinophilic inflammation (Howl JH, Petersen MG. Intestinal mast cell tumor in a cat: presentation as eosinophilic enteritis. J Am Anim Hosp Assoc. 1995;31(6):457-461. doi:10.5326/15473317-31-6-457) and CEE has previously proposed cut-offs to distinguish from LP inflammation with an eosinophilic component (Tucker S, Penninck DG, Keating JH, Webster CR. Clinicopathological and ultrasonographic features of cats with eosinophilic enteritis. J Feline Med Surg. 2014;16(12):950-956. doi:10.1177/1098612X14525385).

Line 152: How the severity of reactive fibroblasts was scored needs to be defined, as this is not part of WSAVA scoring and is currently subjective, as written. Please also cite a basis for how the grades were defined.

Line 153: As above, evaluation of all layers is not possible with endoscopic samples.

Results:

Lines 196-198: Were these statistically assessed/normally distributed -- is mean versus median appropriate?

Lines 200 - 236: It is challenging to assess the validity of the findings and divisions among grades reported in this section. The division of “most affected” sections is misleading, as it appears that 6 cats did not have ileal samples, a highly affected region in previous reports, 3 only had duodenal samples, and 2 only had jejunal samples. It would be more accurate to report the affected region findings in light of how many cats had samples of those regions. As the data is currently presented, it is not possible to know whether regions not sampled were not affected.

The inclusion of cases with endoscopic samples only may substantially bias the percentage of cats with each grade, as a cat with endoscopic samples only would likely not qualify for grade III simply based on a lack of ability to assess. These results should be redone with exclusion of cases without full thickness samples.

Table 6: “Lymphocytes” instead of “Limpocytes”

Immunohistochemistry results (section 3.3): It would be helpful to define subjective qualifications, such as “moderate” for collagen staining throughout. This is more important for this manuscript than a general descriptive study, as the authors are proposing to uses severities of these criteria for disease classification and must be replicable in future studies.

Line 296: “Cytoplasmic” instead of “citoplasmic"

Line 296: Please provide numbers and ideally, statistical analysis, to support the statement of “lower.”

Discussion

Lines 310 - 330: While the authors acknowledge that larger studies and control populations are needed for breed discussion, the statements made overstate the ability of this study to draw any conclusions about breed, as there was no attempt to compare to a general hospital presenting population at the time. Further no statistical analysis was performed (and the study was likely underpowered) to support any statements about different breeds having more or less severe lesions.

330: Please provide statistical analysis to state that cats with more severe grades were older.

Lines 331-338: This grading scheme is inappropriate for endoscopic biopsies, as well as extrapolation for regions included when not all were assessed in many cats. Please revise this section with only inclusion of cases with full thickness results.

Lines 346-348: Please clarify the sentence “Further studies include…protocolized method was applied.” Grammatical editing is needed.

Line 354: The criteria for diagnosis was not described in the methods and should be included.

Lines 375 - 378: Please separate into distinct sentences.

Line 387: Connective tissue?

Line 394: Consider “fibrotic” instead of “fibrosis”

Lines 398-400: This statement becomes problematic when it was unclear how many samples were full thickness and the inherent inability to assess these lesions in endoscopic samples.

Lines 408-410: Please revise this sentence for clarity.

Lines 421-423: Please include a literature citation for this statement.

Lines 424-425: “Thus, proposing…change in the fibroblast” is an incomplete sentence

Line 430: Does “this disease” mean FGESF?

Conclusions

Lines 456-458: It is unclear how this statement fits with the aims or results presented in the study and seems out-of-place when introduced in the conclusions. If relevant to the study objectives and results, please expand in the discussion with relevant literature citations.

Author Response

Reviewer 1

General comments: This manuscript provides some novel information regarding an uncommon cause of feline chronic enteropathies, particularly in the histopath characteristics. However, substantial editing is required to the reported methods and results to clearly evaluate the study’s conclusions. Care should also be taken not to extrapolate possible clinical applications to a retrospective, histopath-based study. Lastly, many comparative statements were made without statistical analysis; statistical analysis should be performed if these conclusions remain in the revised manuscript.

There are multiple areas where grammar requires revision. While I have highlighted some, this should be a general focus of the revision process.

Abstract: The abstract will require revision following other recommended edits to the manuscript (outlined below) to clarify the diagnostic methods applied to the study population.

Introduction:

Line 40: Is “Davidson et al., 2021” the same as reference 9 or a different citation?

We appreciate your comment, and we apologize for the mistake. We have modified the sentence: “The etiopathogenesis of FGESF is not completely understood, partly because the majority of publications deal with single isolated cases [4-9].”

Lines 42-43: While there has been some suggestion that Ragdoll’s are predisposed to FGESF, it is unclear how that relates to the previous sentence about cats in general been predisposed to an eosinophilic response. Evidence of Ragdoll predisposition to other eosinophilic diseases might more clearly connect these thoughts.

We appreciate your comment. We have modified the sentence: “There is the hypothesis that some cats with a genetic predisposition to this lesion develop eosinophilic inflammation in response to certain antigens in the intestinal environment [1]. Moreover, in the study by Linton et al., it was observed that long-haired cats and specifically the Ragdoll breed were overrepresented, suggesting a breed predisposition to this disease [2]. However, further studies concerning the genetic predisposition between affected cats are necessary to reach a conclusion.”

Lines 47-48: Please revise for clarity. Lesions similar to what?

We appreciate your comment. We have modified the sentence: “Some authors have reported similar eosinophilic sclerosing lesions in the subcutaneous tissue and abdomen of cats infected with methicillin-resistant Staphylococcus aureus [5,11].”

Line 58: “Ileum” instead of “ileus”?

We appreciate your comment. We have modified the sentence: “Grossly, FGESF has been roughly described as an intramural, firm, irregular and ulcerated mass affecting the gastrointestinal tract; mainly stomach, pyloric sphincter, ileum, ileocecocolic junction and colon”

Line 65: “Penetrated” instead of “penetrate”

We appreciate your comment. We have modified the sentence: “the mass was attached to the pylorus and affected the proximal duodenum, the bile duct and the pancreas, which penetrated the wall of the mass [9].”

Lines 50-81: Reorganization of these paragraphs could help improve clarity. Intermixed description of examination findings, imaging findings, and cytologic/histopathologic features make this section difficult to follow. Consider grouping all PE findings, then imaging, then pathologic findings together.

We appreciate your comment. We have modified the paragraphs:

“Clinical survey reveals lesions characterized by large, hard, easily palpable mass and most commonly found near the pylorus or ileocaecocolic junction [1,2]. Abdominal palpation usually confirms the presence of one or multiple abdominal masses, sometimes painful, which can be confirmed by complementary imaging techniques (ultrasound, radiology, or computed tomography) [3]. Ultrasonography in association with cytological examination is useful to focus the diagnosis of FGESF, although the definitive diagnosis is made through histopathology [3].”

“Grossly, FGESF has been roughly described as an intramural, firm, irregular and ulcerated mass affecting the gastrointestinal tract; mainly stomach, pyloric sphincter, ileum, ileocaecocolic junction and colon [2,6]. The mesenteric lymph nodes may also be affected, with subsequent mesenteric lymphadenomegaly, suggesting an extension of the inflammatory process [6]. An atypical form of FGESF, characterized by small multifocal firm nodules affecting only the mesentery has been also described [7]. Recently, a case was reported characterized by an extensive intramural cavitated mass in the cranial abdomen: the mass was attached to the pylorus and affected the proximal duodenum, the bile duct and the pancreas, which penetrated the wall of the mass [9]. On rare occasions, the process initiated in the abdomen spreads to the thoracic cavity, probably due to drainage of the abdominal lymph nodes to the sternal lymph node, causing dyspnea [2]. 

Histologically, the key features of FGESF are dense trabeculae of collagen (in some cases resemble as osteoid tissue), interspersed by large spindle-shaped cells (fibroblasts), with accompanying mixed inflammatory cells (composed mainly by eosinophils with fewer lymphocytes, plasma cells and in some cases mast cells) [1,2,5]. Due to its microscopic structure, FGESF can be misdiagnosed as lymphoblastic B-cell lymphoma, osteosarcoma or sclerosing mast cell carcinoma, so the pathologist must be extremely careful to distinguish this disease from a neoplasm [1]. The prognosis is variable and depends on factors such as diagnosis delay, involvement of adjacent organs and response to therapy [3,8].”

Line 75: Consider removing “an”

We appreciate your comment. We have modified the sentence: “Histologically, the key features of FGESF are dense trabeculae of collagen (in some cases resemble as osteoid tissue), interspersed by large spindle-shaped cells (fibroblasts), with accompanying mixed inflammatory cells (composed mainly by eosinophils with fewer lymphocytes, plasma cells and in some cases mast cells) [1,2,5].”

Line 101: Does “this” refer to collagen?

We appreciate your question. Yes, “this” refers to collagen. We have modified the sentence to make it clearer: “It is well known that fibrosis is characterized by an alteration of collagen metabolism that leads to a greater deposition of this extracellular matrix protein in the tissue.”

Lines 101 - 104: Please divide into separate sentences for clarity (“There are several types of…in which the tissue is found”)

We appreciate your comment. We have modified the sentence: “There are several types of collagens, with type I and type III being the main components of the normal stroma.  However, the proportions expressed in the tissues can vary significantly between tissues, both in physiological and pathological conditions [15-17].”

Line 103: Please expand/clarify what is meant by “depending on the state in which the tissue if found.” Does this mean stage of disease progression or stage of fibrogenesis?

We appreciate your comment. We have modified the sentence: “However, the proportions expressed in the tissues can vary significantly between tissues, both in physiological and pathological conditions [15-17].”

Line 110: What is “its” referring to?

We appreciate your question. “its” refers to the growth factors. We have modified the sentence to make it clearer: “However, the prolonged expression of this growth factors during chronic inflammation can induce excessive collagen deposition and eventually fibrosis”.

Lines 126-130: This seems more appropriate for the discussion, as this study is not designed to evaluate progression of disease or whether FGESF is a progression or separate disease process from CEE.

We appreciate your comment. We have removed the sentence from the introduction, and we have moved it to the discussion:

“Histopathology is the gold standard technique for the diagnosis of FGESF. As many aspects of pathogenesis are still not fully understood, we proposed histopathological criteria for both the diagnosis and classification into grades of FGESF lesions in cats, which could serve future studies on disease outcome, prognosis, and therapy.  Moreover, this could be useful, in the future, to differentiate a mild degree (Grade I) of this disease (where the lesion is located exclusively in the mucosal layer) from chronic eosinophilic enteritis (CEE), whose prognosis and evolution is more favorable.”

Materials and Methods

Lines 132-133: It is unclear how many samples were full thickness samples and how many were endoscopic. For the authors’ aims and suggested histopathologic classification, it does not seem appropriate to include cases in which full thickness samples were not obtained. Endoscopic sampling inherently does not allow assessment of all layers of the GI tract, which is an assumption of the grading system. Inclusion of cases with only endoscopic samples is a major limitation to the study’s results and conclusions and should be removed.

We appreciate your comment. Eight samples were endoscopic intestinal biopsies, in which during the endoscopic study, only a thickening of the mucosal layer was observed, without "mass effect". This samples were compatible with an IBD and histologically diagnosed as FGESF grade I in this study (only the mucosal layer is affected, where a continuous band of connective tissue with mild proliferation of fibroblast were observed).

It is necessary to understand that in the routine diagnosis of IBD, where there is no mass present and only the mucosal layer is affected, endoscopic biopsy is usually performed. Therefore, it is very unlikely to obtain complete specimens from animals diagnosed with a grade I. Furthermore, it is interesting to apply this system, proposed in this study, on endoscopic samples to suggest an early differential diagnosis between an initial grade of FGESF and an IBD process like an eosinophilic chronic enteritis with fibrosis, which has a much more favorable prognosis. This has been discussed.

We have modified the text:

“Sixteen feline intestinal samples obtained from well-oriented endoscopic feline intestinal biopsies (n = 8) and full-thickness (n = 8) from two veterinary diagnostic services (Veterinary Teaching Hospital, Faculty of Veterinary Medicine, Complutense University of Madrid; and VetPatólogos diagnostic laboratory) during a 4-year period (2014-2017) diagnosed as FGESF were included in the study. All cases were retrospectively re-evaluated by two experienced veterinary pathologists and two trainee veterinary pathologists with a final diagnosis of FGESF achieved by agreement (Table 1).

The endoscopic biopsies correspond to animals in which, during the endoscopic study, only a thickening of the mucosal layer was observed, without "mass effect". These endoscopic samples were compatible with a chronic eosinophilic enteritis (CEE) and histologically diagnosed as FGESF grade I in this study (discussed later). All animals contributing to this study were sampled from each small intestinal tract (duodenum, jejunum, ileum). Herein, we showed only the intestinal tracts with FGESF-compatible lesions (duodenum n = 14, jejunum n = 13, and ileum n = 10).”

Line 136: The methods should include a description of diagnostic criteria for FGESF. It is unclear whether cases were screened with additional stains for neoplasia or infectious disease or degree of eosinophilic inflammation that was required to categories these cats as having eosinophilic inflammation. This is critical to understand the study population, as MCT have been previously misdiagnosed as eosinophilic inflammation (Howl JH, Petersen MG. Intestinal mast cell tumor in a cat: presentation as eosinophilic enteritis. J Am Anim Hosp Assoc. 1995;31(6):457-461. doi:10.5326/15473317-31-6-457) and CEE has previously proposed cut-offs to distinguish from LP inflammation with an eosinophilic component (Tucker S, Penninck DG, Keating JH, Webster CR. Clinicopathological and ultrasonographic features of cats with eosinophilic enteritis. J Feline Med Surg. 2014;16(12):950-956. doi:10.1177/1098612X14525385).

We appreciate your comment. The samples were screened by two experienced veterinary pathologists and two trainee veterinary pathologists with a final diagnosis of FGESF achieved by agreement, as detailed in the Material and Methods sections. The histopathological features of FGESF have been extensively described (please, see references 1, 2, 3, 5, 8). Humbly, in our cases, there has been no confusion with other types of lesions (neoplastic, infectious).

Material and Methods section has been restructured. Reviewer demands regarding FGESF diagnostic criteria has been achieved by detailing FGESF diagnosis, evaluation and grading criteria:

“The histopathological features that must be fulfilled to diagnose a possible case of FGESF are the following: fibrosis, reactive fibroblast infiltrate, and a mixed inflammatory infiltrate, mainly composed of a moderate to severe eosinophilic component. The samples were reassessed according to the endoscopic, biopsy and histopathological guidelines for the evaluation of gastrointestinal inflammation in companion animals of the World Small Animal Veterinary Association (WSAVA) in a scale of 0-3 (0: absence, 1: mild, 2: moderate and 3: severe) [25]. Furthermore, the eosinophilic inflammation was graded as mild (1: 5–10 eosinophils per 40× field), moderate (2: 10–20 eosinophils per 40× field) or severe (3: eosinophils pre-dominate and are not easily enumerated) based on the study by Tucker et al. on chronic eosinophilic enteritis (CEE) [26]. We have also evaluated the presence of reactive fibroblasts and the presence of foci of necrosis in a scale of 0-3. The degree of reactive fibroblasts infiltrate was graded as mild (1: 10–30 reactive fibroblasts per 40× field), moderate (2: 30-60 reactive fibroblasts per 40× field) or intense (3: >60 reactive fibroblasts). Thus, we evaluated the morphologic and inflammatory parameters detailed in Table 2 in the mucosa, submucosa, muscular and serosa layer.”

Line 152: How the severity of reactive fibroblasts was scored needs to be defined, as this is not part of WSAVA scoring and is currently subjective, as written. Please also cite a basis for how the grades were defined.

We appreciate your comment. We agree with you that the grade classification system should be as objective as possible. That is why we have included in the study the evaluation criteria that we have carried out.

We have included the following text in the material and methods section:

“We have also evaluated the presence of reactive fibroblasts and the presence of foci of necrosis in a scale of 0-3. The degree of reactive fibroblasts infiltrate was graded as mild (1: 10–30 reactive fibroblasts per 40× field), moderate (2: 30-60 reactive fibroblasts per 40× field) or intense (3: >60 reactive fibroblasts).”

Line 153: As above, evaluation of all layers is not possible with endoscopic samples.

We appreciate your comment. As explained above, the endoscopic intestinal biopsies only correspond to animals diagnosed with FGESF grade I in this study, in which only the mucosal layer is affected. During the endoscopic study, no mass was observed, which is one of the main characteristics of intermediate and advanced grades of this disease. Only mucosal thickening was observed, therefore it cannot correspond to advanced degrees of FGESF, where more layers would be affected. This has been mentioned in Material and Methods section and also discussed below.

Results:

Lines 196-198: Were these statistically assessed/normally distributed -- is mean versus median appropriate?

We appreciate your comment. We have included the mean and the median of all groups:

“The mean age of the cats included in the study was 8 years and the median was 6 years (range: 2-16 years). The mean age of the cats included in grade I was 7 years and the median was 5 years (range: 2-13 years); in grade II the mean age was 8 years and the median was 6 years (range: 3-15 years); and, in grade III the mean age was 12 years and the median was 12 years (range: 8-16 years).”

Lines 200 - 236: It is challenging to assess the validity of the findings and divisions among grades reported in this section. The division of “most affected” sections is misleading, as it appears that 6 cats did not have ileal samples, a highly affected region in previous reports, 3 only had duodenal samples, and 2 only had jejunal samples. It would be more accurate to report the affected region findings in light of how many cats had samples of those regions. As the data is currently presented, it is not possible to know whether regions not sampled were not affected.

We appreciate your comment. We regret not having correctly explained the sampling process and the selection of the affected intestinal tracts. We assure you that all animals contributing to this study were sampled from each intestinal tract. However, in tables 1, 5 and 6 only the affected intestinal tracts were included, the rest of the areas were evaluated and did not suffer any type of alteration to be considered in this study, therefore they were omitted in the results.

We have modified the material and methods section, with a better explanation, in order to make it more understandable to the reader: “All animals contributing to this study were sampled from each small intestinal tract (duodenum, jejunum, ileum). Herein, we showed only the intestinal tracts with FGESF-compatible lesions (duodenum n = 14, jejunum n = 13, and ileum n = 10).”

The inclusion of cases with endoscopic samples only may substantially bias the percentage of cats with each grade, as a cat with endoscopic samples only would likely not qualify for grade III simply based on a lack of ability to assess. These results should be redone with exclusion of cases without full thickness samples.

We appreciate your comment. As explained above, the endoscopic intestinal biopsies only correspond to animals diagnosed with FGESF grade I in this study, in which only the mucosal layer is affected. During the endoscopic study, no mass was observed, which is one of the main characteristics of intermediate and advanced grades of this disease. Only mucosal thickening was observed, therefore it cannot correspond to advanced degrees of FGESF, where more layers would be affected. It is for this reason that endoscopic sampling does not affect the results regarding the involvement of layers in FGESF grades II and III. Anyway this has been added in the Material and Methods and in the Discussion section.

Table 6: “Lymphocytes” instead of “Limpocytes”

 We appreciate your comment. We have corrected the word.

Immunohistochemistry results (section 3.3): It would be helpful to define subjective qualifications, such as “moderate” for collagen staining throughout. This is more important for this manuscript than a general descriptive study, as the authors are proposing to uses severities of these criteria for disease classification and must be replicable in future studies.

We appreciate your comment. We agree with you that the study of degrees of severity should be as objective as possible. That is why we have included in the study the evaluation criteria that we have carried out.

We have included the following text in the material and methods section:

“Type I and III collagen immunoreaction was assessed as follows: fibrotic tissue was evaluated in 5 adjacent non-overlapping fields under low power field (LPF) magnification (40×), and percentage of immunostained fibrotic tissue was determined. A score from 0 to 3 was assigned to each sample: grade 1 (mild): 0–30% immunopositive fibrotic tissue; grade 2 (moderate): 30–60% immunopositive fibrotic tissue; grade 3 (intense): more than 60% immunopositive fibrotic tissue.”

“Fibronectin immunoreaction was assessed as follows: 100 fibroblasts were counted in 5 adjacent, non-overlapping fields under a high-power field (HPF) (400×) (500 cells in total), and immunostained cells were expressed as a percentage of the cell count. TGF-β1 immunoreaction was assessed as follows: 100 fibroblasts/macrophages were counted in 5 adjacent, non-overlapping fields under a high-power field (HPF) (400×) (500 cells in total), and immunostained cells were expressed as a percentage of the cell count. A score from 0 to 3 was assigned to each sample: grade 1 (mild): 0–30% immunopositive cells; grade 2 (moderate): 30–60% immunopositive cells; grade 3 (intense): more than 60% immunopositive cells.”

Line 296: “Cytoplasmic” instead of “citoplasmic"

We appreciate your comment. We have corrected the sentence: “There was TGF-β1 cytoplasmic immunostaining in fibroblasts and macrophages in a lower number of cells compared to the grade II (Fig. 3).”

Line 296: Please provide numbers and ideally, statistical analysis, to support the statement of “lower.”

We appreciate your comment. As explained above, we have included the evaluation criteria for IHQ grading immunoexpression (mild, moderate, intense), explained in the material and methods section.

We have modified the sentence according the new data entered: “There was moderate TGF-β1 cytoplasmic immunoexpression in fibroblasts and macrophages (Fig. 3).”

Lines 310 - 330: While the authors acknowledge that larger studies and control populations are needed for breed discussion, the statements made overstate the ability of this study to draw any conclusions about breed, as there was no attempt to compare to a general hospital presenting population at the time. Further no statistical analysis was performed (and the study was likely underpowered) to support any statements about different breeds having more or less severe lesions.

We appreciate your comment. We have modified the sentence: “In Linton et al. study was observed that long-haired cats and specifically the Ragdoll breed were overrepresented, suggesting a breed predisposition to this disease [2]. Here, we reported European Shorthair as the commonly affected breed during a 4-year period. However, in this study, as in Craig's research, no racial predisposition was observed, since the European shorthair cat is representative of the feline population [1]. This suggests that breed distribution may be influenced by geographical location, given the trends in breeds among owners in different countries. Therefore, for an evaluation of a breed predisposition and breed-related severity of lesions wider international multicentric studies are required.”

330: Please provide statistical analysis to state that cats with more severe grades were older.

We appreciate your comment. Due to the small number of samples with FGESF Grade III, the statistical result is not significant. Therefore, we have modified the following sentence:

“Interestingly, our cases with more advanced lesions showed a higher age range (8-16 years). However, a larger population is required to confirm, by statistical study, that there is a relationship between the development of advanced degrees of FGESF and older ages.”

Lines 331-338: This grading scheme is inappropriate for endoscopic biopsies, as well as extrapolation for regions included when not all were assessed in many cats. Please revise this section with only inclusion of cases with full thickness results.

We appreciate your comment. We regret not having correctly explained the sampling process and the selection of the affected intestinal tracts. We assure you that all animals contributing to this study were sampled from each small intestinal tract. However, in tables 1, 5 and 6 only the affected intestinal tracts were included, the rest of the areas were evaluated and did not suffer any type of alteration to be considered in this study, therefore they were omitted in the results tables.

We have modified the material and methods section, with a better explanation, in order to make it more understandable to the reader: “All animals contributing to this study were sampled from each small intestinal tract (duodenum, jejunum, ileum). Herein, we showed only the intestinal tracts with FGESF-compatible lesions (duodenum n = 14, jejunum n = 13, and ileum n = 10).”

Lines 346-348: Please clarify the sentence “Further studies include…protocolized method was applied.” Grammatical editing is needed.

We appreciate your comment. We have modified the sentence: “Further studies should include increased number of cases applying homogeneous evaluation criteria. For that purpose, the histopathological study of the different sections of the intestine should be based on the grading system described here to determine if there is an extension of the fibrotic and inflammatory process that could correlate with an initial grade of the FGESF.”

Line 354: The criteria for diagnosis was not described in the methods and should be included.

We appreciate your comment. We have included the following text in the material and methods section:

“The histopathological features that must be fulfilled to diagnose a possible case of FGESF are the following: fibrosis, reactive fibroblast infiltrate, and a mixed inflammatory infiltrate, mainly composed of a moderate to severe eosinophilic component based on previously reported cases of FGESF [1-3,5,8].”

Lines 375 - 378: Please separate into distinct sentences.

Done as requested.

Line 387: Connective tissue?

We appreciate your question, and we apologize for the mistake. We have modified the sentence: “In some cases of fibrosis, there was a slight staining in the loose connective tissue due to the TGF-β1 protein binding to cellular membranes and the ECM located in the lamina propria.”

Line 394: Consider “fibrotic” instead of “fibrosis”

We appreciate your comment. We have modified the word.

Lines 398-400: This statement becomes problematic when it was unclear how many samples were full thickness and the inherent inability to assess these lesions in endoscopic samples.

We appreciate your question, and we apologize for the mistake. There are no samples taken by endoscopic biopsy in cases of FGESF Grade II and III. In these advanced cases, where a "mass effect" is observed, excisional biopsy has always been performed. As explained above, the endoscopic intestinal biopsies only correspond to animals diagnosed with FGESF grade I in this study, in which only the mucosal layer is affected. During the endoscopic study, no mass was observed, which is one of the main characteristics of intermediate and advanced grades of this disease. Only mucosal thickening was observed, therefore it cannot correspond to advanced degrees of FGESF, where more layers would be affected.

We have modified the following sentence:

“In intermediate grades of FGESF (II), we described two or more intestinal layers affected. The hallmark finding in this grade was the marked presence of activated fibroblast population located between bands of connective tissue in the lamina propria and submucosa and, less commonly, in the muscularis.”

Lines 408-410: Please revise this sentence for clarity.

We appreciate your comment. We have modified the sentence: “This may be related to the moderate inflammatory infiltrate present, stimulating TGF-β1 expression [15,17,21].”

Lines 421-423: Please include a literature citation for this statement.

We appreciate your comment. We have modified the sentence and added a literature citation: “It has been shown that the alteration of the proportion of type III:I collagen and the high expression of the protein and mRNA of TGF-β1 and IGF-1 coincide with the presence of an inflammatory infiltrate in the intestine of human individuals suffering from IBD [15]”

Lines 424-425: “Thus, proposing…change in the fibroblast” is an incomplete sentence

We appreciate your comment, and we apologize for the mistake. We have modified the sentence: “Thus, it is proposed that the inflammatory infiltrate is the trigger for the increase in TGF-β1 and IGF-1 expression, inducing a phenotypic change in the fibroblast.”  

Line 430: Does “this disease” mean FGESF?

We appreciate your question. Yes, “this disease” means FGESF. We have modified the sentence: “These results suggest the need to differentiate between FGESF and IBD, since previous studies agree that the presence of eosinophils plays a very important role in the development of intestinal fibrosis in FGESF”

Conclusions

Lines 456-458: It is unclear how this statement fits with the aims or results presented in the study and seems out-of-place when introduced in the conclusions. If relevant to the study objectives and results, please expand in the discussion with relevant literature citations.

We appreciate your comment, and we apologize it is not understandable or out-of-place for the reader. We have removed this sentence from the conclusion to avoid misunderstandings.

Reviewer 2 Report

Dear authors,

I consider your work to be a very important contribution to the study of this clinical situation.

I suggest using a title closer to the core of the work.

For instance,...

Feline gastrointestinal eosinophilic sclerosing fibroplasia - proposal of a new classification system involving histological and immunohistochemical evaluation parameters.

Introduction

Line 35 - None of the references can be considered recent, the oldest is from 2009 and the most recent from 2015, so the phrase "has recently been described" should be reconsidered. The designation of the disease was already proposed in 2009.

Line 40 - Davidson et al., 2021]. ?

Line 40 - There is the hypothesis that some cats (The cited reference, which I consulted, refers to a hypothesis involving some genetically predisposed cats.)    -  "We hypothesize that cats with a genetic predisposition to this lesion develop eosinophilic inflammation in response to the introduction of bacteria (or other antigens) into the intestinal wall, perhaps by a foreign body or ulceration."

Line 49 - Staphylococcus aureus (please in italics)

Lines 53 to 55 - please articulate these two sentences:

"The main clinical manifestations in affected cats include vomiting and chronic diarrhea, weight loss, constipation, hyporexia and pale mucous membranes [1,2]."

as for instance.... "The main clinical manifestations in affected cats include vomiting and chronic diarrhea, hyporexia, weight loss, pale mucous membranes and  constipation as intestinal fibrosis causes stenosis and signs of obstruction. "

Lines 57 to 59 - please describe according to the normal anatomical sequence

Line 95 - and can lead to the proliferation of fibroblasts and extracellular matrix proteins deposition. since EMP does not proliferate.

Lines 96 to 99 - Prednisone inhibits the production of leukotrienes by interrupting the arachidonic acid route and therefore cats affected by FGESF and treated with prednisone showed the longest survival times

The sentence needs to be punctuated and the reference added.

Results

Lines 204 to 205 - In 6/8 cases (75.0%), the disease affected the whole of the small intestine, in 1/8 cases (12.5%) there were lesions in the duodenum and jejunum and in 1/8 cases (12.5%) just in the duodenum.

Line 204 - with chronic eosinophilic enteritis (CEE).

Line 265 - Regarding figure 1, the images must be larger and the changes described in the legend must be properly marked on the image, so that readers can identify them. Please add scale bars to all the histological figures.

Discussion

Line  424 - in TGF-β1

Line 429 - that the presence of eosinophils

Conclusions

Line 452 - TGF-β1

Author Response

Reviewer 2

Dear authors,

I consider your work to be a very important contribution to the study of this clinical situation.

I suggest using a title closer to the core of the work.

For instance,

Feline gastrointestinal eosinophilic sclerosing fibroplasia - proposal of a new classification system involving histological and immunohistochemical evaluation parameters.

We appreciate the Reviewer suggestions. Based on the modifications proposed by both reviewers, we believe that the most appropriate tittle would be: “Feline gastrointestinal eosinophilic sclerosing fibroplasia – TGF-β1 and extracellular matrix proteins immunoexpression”. We hope and anticipate it contains the main idea of the manuscript. Anyway, we are open to the reviewer opinion.

Introduction

Line 35 - None of the references can be considered recent, the oldest is from 2009 and the most recent from 2015, so the phrase "has recently been described" should be reconsidered. The designation of the disease was already proposed in 2009.

We appreciate your comment. We have modified the sentence: “Feline gastrointestinal eosinophilic sclerosing fibroplasia (FGESF) has been described as part of the inflammatory diseases with eosinophilic component…”

Line 40 - Davidson et al., 2021]. ?

We appreciate your comment, and we apologize for the mistake. We have modified the sentence: “The etiopathogenesis of FGESF is not completely understood, partly because the majority of publications deal with single isolated cases [4-9].”

Line 40 - There is the hypothesis that some cats (The cited reference, which I consulted, refers to a hypothesis involving some genetically predisposed cats.)    -  "We hypothesize that cats with a genetic predisposition to this lesion develop eosinophilic inflammation in response to the introduction of bacteria (or other antigens) into the intestinal wall, perhaps by a foreign body or ulceration."

We appreciate your comment. We have modified the sentence: “There is the hypothesis that some cats with a genetic predisposition to this lesion develop eosinophilic inflammation in response to certain antigens in the intestinal environment [1]. Moreover, in the study by Linton et al., it was observed that long-haired cats and specifically the Ragdoll breed were overrepresented, suggesting a breed predisposition to this disease [2]. However, further studies concerning the genetic relationship between affected cats are necessary to reach a conclusion.”

Line 49 - Staphylococcus aureus (please in italics)

We appreciate your comment. We have modified the format: “Staphylococcus aureus”.

Lines 53 to 55 - please articulate these two sentences:

"The main clinical manifestations in affected cats include vomiting and chronic diarrhea, weight loss, constipation, hyporexia and pale mucous membranes [1,2]."

as for instance.... "The main clinical manifestations in affected cats include vomiting and chronic diarrhea, hyporexia, weight loss, pale mucous membranes and constipation as intestinal fibrosis causes stenosis and signs of obstruction. "

We appreciate your comment. We have modified the sentence: “The main clinical manifestations in affected cats include weight loss, hyporexia, pale mucous membranes, vomiting, chronic diarrhea, and on other occasions, constipation, as intestinal fibrosis causes stenosis and signs of obstruction.”

Lines 57 to 59 - please describe according to the normal anatomical sequence

We appreciate your comment. We have modified the sentence: “Grossly, FGESF has been roughly described as an intramural, firm, irregular and ulcerated mass affecting the gastrointestinal tract; mainly stomach, pyloric sphincter, ileum, ileocecocolic union and colon [2,6].”.

Line 95 - and can lead to the proliferation of fibroblasts and extracellular matrix proteins deposition. since EMP does not proliferate.

We appreciate your comment. We have modified the sentence: “Fibrogenic mediators, such as transforming growth factor β (TGF-β) and interleukin-1β (IL-1β), are produced by the activation of eosinophils and can lead to the proliferation of fibroblasts and extracellular matrix proteins deposition.”

Lines 96 to 99 - Prednisone inhibits the production of leukotrienes by interrupting the arachidonic acid route and therefore cats affected by FGESF and treated with prednisone showed the longest survival times

The sentence needs to be punctuated and the reference added.

We appreciate your comment. We have modified the sentence: “Prednisone inhibits the production of leukotrienes by interrupting the arachidonic acid route. Therefore, cats affected by FGESF and treated with prednisone showed the longest survival times [1].”.

Results

Lines 204 to 205 - In 6/8 cases (75.0%), the disease affected the whole of the small intestine, in 1/8 cases (12.5%) there were lesions in the duodenum and jejunum and in 1/8 cases (12.5%) just in the duodenum.

We appreciate your comment. We have modified the sentence: “In 6/8 cases (75.0%), the disease affected the whole of the small intestine, in 1/8 cases (12.5%) there were lesions in the duodenum and jejunum and in 1/8 cases (12.5%) just in the duodenum.”.

Line 204 - with chronic eosinophilic enteritis (CEE).

We appreciate your comment. We have modified the sentence: “Half of cases (8/16, 50.0%) were classified as grade I, which is histologically compatible with chronic eosinophilic enteritis (CEE)“.

Line 265 - Regarding figure 1, the images must be larger and the changes described in the legend must be properly marked on the image, so that readers can identify them. Please add scale bars to all the histological figures.

We appreciate your comment. The size of the images is subject to the format of the template and Author Guidelines. We have modified the figures by adding the scale bars, which is defined in the figure legend. Also we have added arrows and arrowheads to mark the lesions described in the figure legends to make it more understandable to the reader.

Discussion

Line  424 - in TGF-β1

We appreciate your comment. We have modified the sentence: “Thus, proposing that the inflammatory infiltrate is the trigger for the increase in TGF- β1 and IGF-1 expression, inducing a phenotypic change in the fibroblast.”.

Line 429 - that the presence of eosinophils

We appreciate your comment. We have modified the sentence: “These results suggest the need to differentiate between FGESF and IBD, since previous studies agree that the presence of eosinophils plays a very important role in the development of intestinal fibrosis in FGESF [1,2,5].”.

Conclusions

Line 452 - TGF-β1

We appreciate your comment. We have modified the sentence: “In addition, our results also suggest an important role of TGF-β1, triggering fibroblastic proliferation and collagen and fibronectin deposition.”

Round 2

Reviewer 1 Report

The reviewer would like to thank the authors for their efforts to clarify the study design and results; this has much-improved the quality of the manuscript. There are still some grammatical and spelling errors that I trust will be addressed during the editing process. A few other, minor comments are below:

Materials and methods:

Line 142: Should “endoscopic” be “ultrasonographic”?

Line 146: It is considered highly unlikely that jejunal samples were obtained in all cats with endoscopic biopsies, a limitation of an endoscopic approach. Please clarify or state how it was possible to determine that the jejunum was reached.

Discussion:

While the methods and results have been more clearly outlined to define the population of included cats, there should still be discussion of the limitation of including cats with endoscopic-only biopsies. While the authors are including these cats as possible, grade I FGESF, there is currently not evidence to support the inflammation would progress past “eosinophilic enteritis” to the more common mass-like presentation that the authors use to define the disease in the introduction. While the mechanism for progression is interesting, the limitation in grouping these cats as FGESF should be discussed.

Further, while the reviewer respects the decision not to pursue full thickness biopsies on cats without an ultrasound (suspecting that there was a typo in the method) mass-effect, it cannot be concluded that the histopath lesions are limited to visible ultrasound lesions. It is well-known that cats with severe idiopathic inflammatory lesions and lymphoma on histopathology can have completely normal ultrasounds. Therefore, the possibility that more extensive lesions and possible mis-classification as a lower grade, should be discussed.

Author Response

Reviewer 1

Comments and Suggestions for Authors

The reviewer would like to thank the authors for their efforts to clarify the study design and results; this has much-improved the quality of the manuscript. There are still some grammatical and spelling errors that I trust will be addressed during the editing process. A few other, minor comments are below:

Thank you for your comments. We greatly appreciate the effort you have made during your review. It has significantly improved the quality of our manuscript.

Materials and methods:

Line 142: Should “endoscopic” be “ultrasonographic”?

We appreciate your comment.  We have modified the following sentence according your suggestion: “The endoscopic biopsies correspond to animals in which, during the ultrasonographic and endoscopic study, only a thickening of the mucosal layer was observed, without "mass effect".

Line 146: It is considered highly unlikely that jejunal samples were obtained in all cats with endoscopic biopsies, a limitation of an endoscopic approach. Please clarify or state how it was possible to determine that the jejunum was reached.

We appreciate your comment.  It is true that endoscopic sampling in the jejunum is less frequent, but it is possible to perform it, at least in smaller animals, such as cats but it depend on the characteristic of the endoscopy.

This fact has been confirmed and published: “After appropriate insufflation the small intestinal lumen is usually well visualized during enteroscopy. The descending duodenum can be examined in all animals in which the endoscope can be advanced through the pylorus, and the jejunum can often be reached in cats and small dogs.”

Endoscopic Examination of the Small Intestine

Todd R. Tams, Craig B. Webb, in Small Animal Endoscopy (Third Edition), 2011

https://doi.org/10.1016/B978-0-323-05578-9.10005-1

We have modified the following sentence in the material and methods section:

“All animals contributing to this study were sampled from each small intestinal tract (duodenum, proximal jejunum, ileum).”

Discussion:

While the methods and results have been more clearly outlined to define the population of included cats, there should still be discussion of the limitation of including cats with endoscopic-only biopsies. While the authors are including these cats as possible, grade I FGESF, there is currently not evidence to support the inflammation would progress past “eosinophilic enteritis” to the more common mass-like presentation that the authors use to define the disease in the introduction. While the mechanism for progression is interesting, the limitation in grouping these cats as FGESF should be discussed.

Further, while the reviewer respects the decision not to pursue full thickness biopsies on cats without an ultrasound (suspecting that there was a typo in the method) mass-effect, it cannot be concluded that the histopath lesions are limited to visible ultrasound lesions. It is well-known that cats with severe idiopathic inflammatory lesions and lymphoma on histopathology can have completely normal ultrasounds. Therefore, the possibility that more extensive lesions and possible mis-classification as a lower grade, should be discussed.

Thank you very much for your comments. The protocols established in veterinary clinics and/or hospitals are very varied. In our case, we first perform blood biochemistry and serological studies to rule out systemic or infectious disease (FeLV and FIV), 3-day serial coprological studies, ultrasound study and finally endoscopy. Probably, the non-invasive and economic factors do not make the laparoscopic study with excisional biopsies a routine technique. Of course, we are aware of the topographical and histopathological limitations of our protocol (reaching the proximal jejunum and not seeing beyond the mucosal layers and the superficial area of the submucosa).

According to the experience of our clinicians and sonographers, it is rare to find a normal sonographic image in animals with severe IBD or lymphoma; at least a thickening of the wall is seen in animals with IBD and in the case of lymphoma, this thickening is accompanied by a loss of layers. However, this mural thickening is not characteristic of any particular entity. It does, however, make gastrointestinal endoscopic study indicated. The collection of numerous samples (10 to 12 samples) in different locations increases the chances of determining the pathologic process that is present in the stomach and/or intestine. Ultrasonographically, these type I lesions showed diffuse mural thickening, there was no “mass effect”. The formation of a continuous band of collagen above the muscularis of the mucosa has been a finding appreciated in the three grades, although it was more typical of grade I FGESF, becoming more extensive in grade II FGESF with involvement also of the superficial lamina propria and the base of the villi, and extending to other layers in grade III FGESF. It is of vital importance to understand that the "mass effect" is caused by the formation of dense collagen trabeculae, with a high proportion of type I collagen (assessed by IHC). The IHC study in grade I demonstrates a high expression of collagen III in a reticular type fibrosis (early fibrosis) in the lamina propria, which cannot cause a mass effect, only a thickening of the mucosa. The laparoscopic procedure is more indicated in those animals that ultrasonographically show a gastrointestinal mass effect, sometimes associated with lymphadenomegaly. This makes it possible to identify a neoplastic process, such as carcinoma or lymphoma, as well as to rule out a FGESF.

We have included a brief explanation of the presence or absence of the mass effect in each type of grade described above, in the discussion.

As you rightly comment, we cannot 100% confirm this evolution from grade I to grade II. However, in the manuscript we have exposed evident facts suggesting this progression. And as we previously explained, the expression of lesions characteristic of grade I are also seen in more advanced grades, suggesting a progression to deeper layers. In addition, we only grouped CEE cases and due to their particular fibrotic lesions, we suggest a possible grade I FGESF, and we reinforce this theory by IHC evaluating extracellular matrix proteins and TGF.

Moreover, we are currently following up these grade I animals in order to determine that this lesion could be key to identify the onset of feline gastrointestinal eosinophilic sclerosing eosinophilic sclerosing fibroplasia. However, as you can understand, this is a long lasting procedure and sampling is not dependent on us.

Following your advice, we have made a modification to the discussion by commenting on the limitations of the study:

“The limitations of performing histologic study of grade I FGESF only on endoscopic samples should be mentioned. Although during the ultrasound and endoscopic studies no lesion with "mass effect" was observed, we cannot confirm that there is no involvement of deeper layers (submucosal, muscular, serous). Therefore, further studies with full-thickness samples would be recommended to confirm our hypothesis."